# Cultivable Microbiome Approach Applied to Cervical Cancer Exploration

**DOI:** 10.3390/cancers16020314

**Published:** 2024-01-11

**Authors:** Irma Berenice Mulato-Briones, Ismael Olan Rodriguez-Ildefonso, Julián Antonio Jiménez-Tenorio, Patricia Isidra Cauich-Sánchez, María del Socorro Méndez-Tovar, Gerardo Aparicio-Ozores, María Yicel Bautista-Hernández, Juan Francisco González-Parra, Jesús Cruz-Hernández, Ricardo López-Romero, Teresita María del Rosario Rojas-Sánchez, Raúl García-Palacios, Ónix Garay-Villar, Teresa Apresa-García, Juan López-Esparza, Daniel Marrero, Juan Arturo Castelán-Vega, Alicia Jiménez-Alberto, Mauricio Salcedo, Rosa María Ribas-Aparicio

**Affiliations:** 1Laboratorio de Producción y Control de Biológicos, Departamento de Microbiología, Escuela Nacional de Ciencias Biológicas, Instituto Politécnico Nacional, Mexico City 11350, Mexico; imulatob@ipn.mx (I.B.M.-B.); irodriguezi0900@alumno.ipn.mx (I.O.R.-I.); jjimeneztn@ipn.mx (J.A.J.-T.); jcastelv@ipn.mx (J.A.C.-V.); ajimeneza@ipn.mx (A.J.-A.); 2Unidad de Investigación en Biomedicina y Oncología Genómica (UIBOG), del Hospital de Gineco Pediatría No. 3A, del Instituto Mexicano del Seguro Social (IMSS), Mexico City 07300, Mexico; ricardolopez007@gmail.com; 3Laboratorio de Biotecnología Molecular y Farmacéutica, Departamento de Microbiología, Escuela Nacional de Ciencias Biológicas, Instituto Politécnico Nacional, Mexico City 11350, Mexico; 4Laboratorio de Bacteriología Médica, Departamento de Microbiología, Escuela Nacional de Ciencias Biológicas, Instituto Politécnico Nacional, Mexico City 11350, Mexico; pcauich@ipn.mx (P.I.C.-S.); gaparico@ipn.mx (G.A.-O.); 5Laboratorio de Bacteriología Clínica, Hospital General, Centro Médico Nacional “La Raza”, IMSS, Mexico City 02990, Mexico; socomd_1@hotmail.com; 6Unidad de Radiología, Hospital General de México “Dr. Eduardo Liceaga”, Secretaría de Salud, Mexico City 07300, Mexico; maria.bautista@salud.gob.mx (M.Y.B.-H.); jfgparra@yahoo.com.mx (J.F.G.-P.); maximus.nietzsche@gmail.com (J.C.-H.); 7Clínica de la Mujer, Hospital Privado, Mexico City 06760, Mexico; tererojass@yahoo.com.mx; 8Clínica de Atención a la Mujer, Mexico City 03310, Mexico; teresa_t_a@yahoo.com.mx; 9Departamento de Braquiterapia, Hospital de Oncología, Centro Médico Nacional Siglo XXI, IMSS (DBHOCMN-IMSS), Mexico City 07300, Mexico; ongavi@yahoo.com; 10Unidad de Investigación Médica en Enfermedades Oncológicas, Hospital de Oncología, Centro Médico Nacional Siglo XXI, IMSS, Mexico City 07300, Mexico; tapresag@gmail.com; 11Laboratorio de H109, Academia de Microbiología, Instituto de Ciencias Biomédicas, Universidad Autónoma de Ciudad Juárez, Ciudad Juárez 32310, Mexico; maosal89@yahoo.com; 12Unidad de Investigación Médica en Enfermedades Endocrinas, Hospital de Especialidades, Centro Médico Nacional Siglo XXI, IMSS, Mexico City 07300, Mexico; dan.mar57@gmail.com

**Keywords:** microbiology, aerobic and anaerobic cultures, cervical epithelium, cervical cancer, culturomics

## Abstract

**Simple Summary:**

The isolation and identification of microbes from cancerous lesions are a technical challenges but can contribute to diagnosing the etiology of cervical neoplastic disease. Massive sequencing techniques have facilitated the characterization of microbial communities in the cervix, and the culturomics approach complements the understanding of the mechanistic interactions between microbes and their host. This pilot work aimed to cultivate the cervical microbiota of women with cervical cancer and women without cervical cancer, using media culture conditions available in a bacteriology laboratory coupled with high-performance identification techniques, and to characterize the differences in the microbial composition between both groups. The finding led to a graphical model of hypothetical interactions, indicating changes in cervical microbiota composition, and highlighting some groups of microorganisms relevant to the micro-ecosystem interaction dynamics during each epithelium stage. Thus, the present data provide functional aspects related to living microorganisms that had been scarce in the Mexican population.

**Abstract:**

Traditional microbiological methodology is valuable and essential for microbiota composition description and microbe role assignations at different anatomical sites, including cervical and vaginal tissues; that, combined with molecular biology strategies and modern identification approaches, could give a better perspective of the microbiome under different circumstances. This pilot work aimed to describe the differences in microbiota composition in non-cancer women and women with cervical cancer through a culturomics approach combining culture techniques with Vitek mass spectrometry and 16S rDNA sequencing. To determine the possible differences, diverse statistical, diversity, and multivariate analyses were applied; the results indicated a different microbiota composition between non-cancer women and cervical cancer patients. The Firmicutes phylum dominated the non-cancer (NC) group, whereas the cervical cancer (CC) group was characterized by the predominance of Firmicutes and Proteobacteria phyla; there was a depletion of lactic acid bacteria, an increase in the diversity of anaerobes, and opportunistic and non-typical human microbiota isolates were present. In this context, we hypothesize and propose a model in which microbial composition and dynamics may be essential for maintaining the balance in the cervical microenvironment or can be pro-oncogenesis microenvironmental mediators in a process called *Ying-Yang* or have a protagonist/antagonist microbiota role.

## 1. Introduction

The knowledge of bacterial communities in the female genital tract has been widely studied because this anatomical site represents a dynamic and very complex system with continuous changes in bacterial colonization, which is partially well-known [1]. It is suggested that bacteria could be an essential risk factor in carcinogenesis, such as *Helicobacter pylori* and its gastric cancer relationship [2] or *Fusobacterium nucleatum* and colorectal carcinoma [3]. For Cervical Cancer (CC), the Human Papillomavirus (HPV) infection is required and is the main risk factor, but it is not enough to explain the development of all CC cases [4]. Several studies have consistently demonstrated that the depletion species of *Lactobacillus*, highly diverse vaginal microbiota, and the presence of oncogenic HPV are risk factors related to cancer progression [5,6].

Any bacteria in the cervicovaginal environment, alive or dead, is detected through massive sequencing data analysis and makes a more accurate description of the specific composition within this anatomical site over traditional methods [7]. In contrast, classical bacteriology mainly detects, isolates, and preserves living bacteria from gynecological infections, allowing for the study of their genotype/phenotype, specific characteristics, and community or individual dynamics [8,9]. Currently, culture-traditional methods focus on detecting known pathogens of the vaginal tract, neglecting other bacterial species, with a possible impact on host health [10]. Moreover, due to financial reasons, analyst abilities, and culturing challenges, the isolation and identification of anaerobic bacteria are rare in traditional disease screening [11]. Thus, the culturomics approach, allowing for extensive analysis of the microbial composition despite different metabolisms [12], can lead to the discovery of novel species from human samples, their characterization, and the interaction with other species and their environment [13], providing new perspectives on host–bacteria relationships [14].

Studies of microbial communities in this environment need a combination of metagenomics approaches along with culture-traditional methods and cultured-independent techniques to expand the human microbiota repertoire knowledge [15] and to increase clinically relevant information on cervical microbiota in disease [10].

The present pilot work aims to describe the differences in microbiota composition for Mexican women with or without CC using traditional microbiological techniques combined with modern identification procedures to elucidate the real microbial interactions on the epithelium due to moderate alterations during cervical cancer development.

## 2. Material and Methods

### 2.1. Biological Samples

The present work represents a transversal, exploratory, and pilot microbiology study from cervical smear samples of patients recruited between 2012 and 2015. The sample size was estimated statistically. A total of 50 non-cancer (NC) women aged 20–63 who attended gynecologic examinations at the Reproduction Clinic Center and the Tepeyac Clinical Center were recruited for this study. The patients presented normal cervical smears and HPV-negative lab results. Any abnormal smear was discarded from this study. A total of 49 women with CC, at any age, treatment, stage, and with biopsy-confirmed disease, were referred to this study by oncologists from the Brachytherapy Unit of CMN-SXXI-MMS, CDMX, and Radiology Unit of Hospital General de Mexico, CDMX. The protocol was reviewed and accepted by the Ethic and Research Committee from the IMSS (R-2014-785-019 registry).

All women in the present study signed an informed consent letter before taking the sample, answered a life habits questionnaire, and cleared all their concerns. During the procedure, the process and intended use of the sample were explained and additional concern clarification was completed. Untraceable samples after the swab collection were excluded.

Inclusion criteria comprise women without antibiotic or antifungal treatment until seven days before the sampling, without sexual intercourse until five days before the sampling, without pregnancy until three months, and with previous external genital water cleaning without hygienical products, perfumes, or soaps that can alter the microbiota.

### 2.2. Demographics Data

The risk factors were evaluated for all patients according to the Mexican Official Guide NOM-014-SSA2-1994 [16], including HPV infection. Obesity and overweight were determined by the body mass index (BMI), considering a BMI > 25 as an overweight parameter. Smoking, age, and high-risk sexual behaviors were recovered from the previous life habits questionnaire and their medical histories; for the high-risk sexual behaviors, the onset age of sex life, the number of sexual partners, and persistent sexual infections were considered.

Additionally, the presence of *Lactobacillus* in the NC group and its absence in the CC group was reported; it was considered as the Nugent score and the presence or absence of the morphology genera in the analysis of freshly obtained samples.

### 2.3. Cervical Sampling Procedure

A gynecologist collected cervical exocervix samples using a sterile cytobrush and a sterile speculum in a previously cleaned genital area; all swabs were mechanically collected and transported on the lapse of 30 min after being taken. Three cytobrushes per individual were used: the first for Stuart media transport, the second for the physiological saline solution to wet mount test, and the detection under the microscope of “clue cells” and the protozoan *Trichomonas vaginalis*. Alternatively, the third cytobrush was used for Gram-staining, and each smear stained was evaluated according to the criteria scoring system Nugent [17]. During the lab analysis, samples without cytobrushes’ traceability were discarded, and the samples were coded to guarantee complete anonymity.

### 2.4. Microbiological Culture and Identification Procedure

The Stuart brush was used to culture the samples on plates of 5% blood-agar and gonococcal agar (GC) with poly-enrichment under microaerophilic conditions during 24–48 h at 35 ± 2 °C. Cultures in MacConkey-agar and mannitol salt-agar were incubated in aerobic conditions at 24–48 h at 35 ± 2 °C, and cultures in Biggy agar were maintained at 25 ± 2 °C for 7 days. Columbia 5% blood-agar, blood-agar hemin-menadione, and thioglycolate medium were inoculated and incubated for 48 h at 35 ± 2 °C under anaerobic conditions using a Gas-Pak jar (Oxoid® AnaeroJarTM, Thermo Fisher Scientific, Basingstoke, UK), with a BD BBL^TM^ GasPak^TM^ (BD, Sparks, NV, USA) with a blue methylene strip as a redox indicator.

After the incubation, all the different morphologic colonies present on the plates were recovered, and all isolates were subjected to Cowan and Steel criteria for genus assignments, such as Gram staining, oxidase, and catalase tests. Then, they were subcultured to ensure the strain purity, identification, and conservation using the seed-batch system.

### 2.5. Microbiota Identification by Mass Spectrometry

Automated microbial identification equipment, the VITEK^®^ MS System (bioMérieux, Marcy l’Etoile, France), identified the isolated colonies; the device uses mass spectrometry through Matrix-Assisted Laser Desorption/Ionization-Time of Flight (MALDI-TOF) technology that creates an MS unique spectrum as well as a comprehensive database of clinically relevant species. In addition, VITEK^®^ MS was also used to identify fungal isolates. The identification of each strain was repeated two times. Identities with confidence scores superior to 90% were considered a correct ID; *Escherichia coli* ATCC 8739 was used as the instrument calibrator strain.

### 2.6. Bacterial 16S rDNA Identification

Previously, cell pellets from bacterial culture broths were harvested, then the cells were treated with K proteinase (28 mg/mL) at 55 °C overnight, and bacterial DNA was extracted with 3M ammonium acetate using the salting-out method. The recovered DNA was quantified using NanoDrop ND-1000 (Thermo Fisher Scientific, Waltham, MA, USA) and adjusted to 100 ng/µL per sample for a PCR assay. A 16S rDNA gene PCR amplification using primers 16S Forward/16S Reverse with the Go Tag^®^ Green Master mix kit (Promega, Madison, WI, USA) was performed; then, an initial denaturation of 94 °C for 5 min, followed by 30 PCR reaction cycles at 94 °C for 30 s, annealing at 55 °C for 30 s, and extension at 72 °C for 30 s; at the end, a final extension at 72 °C for 5 min was conducted. PCR products were purified with the Wizard^®^ SV Gel and PCR Clean-Up System kit (Promega, Madison, WI, USA) according to the manufacturer’s procedure. Amplicon 16S rDNA sequencing allowed the strain identification at the species level.

### 2.7. Uncommon Isolate Detection

The search for uncommon microorganisms in the cervical microbiota was carried out according to the bacteria culture traditional procedure already described. All strains with a molecular identification were subjected to a subsequent strict literature review to find data about their frequency on cervical specimens and other non-human ecological niches.

### 2.8. Statistical Analysis

To determine whether the sample is representative, we used a statistical analysis of the sample size following: $n=\left(z^{2}\times p\left(1-p\right)/e^{2}\right)/\left(1+\left(z^{2}\times p\left(1-p\right)/\left(e^{2}N\right)\right)\right)$. For the variables, the 9439 new cases of CC in 2020 in Mexico were considered, using a 95% confidence level and 15% error. The results suggest that 43 women with CC are an adequate sample for our population, so the analysis is solid considering the 49 CC patients and 50 NC.

The *t*-student’s test for two independent samples was calculated to estimate the number of similarities or dissimilarities between cultures for each group of women. All *p*-values correspond to two-tailed tests. A *p* < 0.05 value was considered significant. The statistical analysis was performed using the XLSTAT 2018.1 program.

### 2.9. Diversity Index

The diversity evaluation was performed using the PAST 4.08 program; the alpha diversity index, the diversity profiles comparing NC and CC groups, and the beta diversity index were performed according to the Whittaker index.

### 2.10. Multivariate Analysis

PCA analysis using a Var–Covar matrix and a similarity Hierarchical cluster assay were performed with the PAST 4.08 program.

## 3. Results

### 3.1. Study Population

The general characteristics of the women populations are shown in Table 1. As expected, the NC group was younger than the CC group. The NC group was mainly overweight with >18 years for their first sexual intercourse (FSI), non-smokers, a Nugent score < 3, and HPV negative. In contrast, the CC group was characterized by being >40 years, non-overweight, an early FSI, non-smokers, under treatment, or at the beginning of therapy. Notably, age was associated with NC women compared to the CC group. Regarding BMI, it highlights that NC women were mostly overweight. As expected, the FSI in the CC population was associated with an earlier starting age, whereas the NC population primarily began their sex life after 18 years old. In addition, we observed that the number of sexual partners and smoking have no statistically significant differences.

It was outstanding that the NC group was positive to clue cells detection, but no woman reported discomfort, so these alterations probably went unnoticed, while the CC group was negative for those criteria. It was also observed that there was a noticeable correlation of CC patients with HPV infection.

### 3.2. Microbiota Composition in Non-Cancer Group of Women

Overall, using microbiological methods, including solid plate culture and enrichment broths, and under aerobic, anaerobic, and micro-aerophilic conditions, 338 isolates were obtained: 329 bacteria and 9 fungal strains (Figure 1). The combined bacteria identification techniques provided a reliable strain characterization at the genera and species levels.

Interestingly, the number of isolates from the CC group was greater than the NC group, but no significant difference between these two groups was noted (Figure 2A). Then, the data were analyzed to know the phyla enrichment in the samples. For the 50 cervical NC samples, 162 isolates were phenotypically identified, yielding 3.2 isolates/woman, but almost 50% of the cases showed a sole isolate. Firmicutes phylum comprised 64.8% of the isolates, followed by 22.8% for Actinobacteria, 9.3% for Proteobacteria, 2.5% for Bacteroidetes, and 0.6% for Fusobacteria; no yeast or parasite were detected (Figure 2B). The Firmicutes phylum percentage was statistically significant (*p* < 0.05, Fisher’s exact test).

After analysis of the isolates, the data show at least 4 groups: (1) a small group only for *Lactobacillus*; (2) *Lactobacillus* spp. plus *Staphylococcus*; (3) *Staphylococcus* spp. plus *Streptococcus*, and (4) mainly Proteobacteria. The most representative species found of Lactobacilli were *Lactobacillus jensenii* and *Lactobacillus crispatus*. *Staphylococcus epidermidis* represented the *Staphylococcus* genus, followed on frequency by *Streptococcus anginosus*, and finally, *Enterococcus faecalis* and *E. coli* were observed.

### 3.3. Microbiota Composition in Cervical Cancer Group of Women

In contrast, from the 49 cervical scrapes from CC women, 176 isolates were obtained with a yield of 3.9 isolates/patient. Firmicutes was found in 47.7% of them, followed by Proteobacteria in 25.6%, Actinobacteria in 14.8%, Bacteroidetes in 6.3%, Ascomycota in 5.1%, and Fusobacteria in 0.6% (Figure 2B). It was noticeable that a Proteobacteria and Ascomycota enrichment that was statistically significant (*p* < 0.05, Fisher’s exact test) was observed (Figure 2B).

The most overrepresented genera were *Streptococcus* and *Staphylococcus*, and in a low frequency, the *Enterococcus, Paenobacillus,* and *Gemella,* as well as the strictly anaerobic bacteria *Clostridium, Anaerococcus, Finegoldia, Parvinomas*, and *Veillonella*. For Actinobacteria phyla, the most representative was *Corynebacterium,* followed by *Actinomyces* and *Gardnerella*; *Arthrobacter*, *Cutibacterium,* and *Trueperella,* were grown as unique isolates. High diversity for the Proteobacteria phyla was observed, the most prominent being the Enterobacteriaceae, and among them, *E. coli* (6 isolates) followed by *Acinetobacter*, *Campylobacter*, *Citrobacter*, *Enterobacter*, *Morganella*, *Obesumbacterium*, *Oligella*, *Proteus*, *Pseudomonas*, and *Serratia*; for the Bacteroidetes and Fusobacterium phyla there were *Bacteroides* and *Prevotella,* and *Fusobacterium nucleatum*, respectively (Figure 1).

Interestingly, a slight diversity increase for aerobic and anaerobic bacteria for the advanced clinical stages was noted, obtaining pure isolates of *Clostridium*, *Escherichia*, and *Corynebacterium* for these patients.

According to the early cancer clinical stage data, the isolates were mainly *Corynebacterium*, *Streptococcus*, *Escherichia*, and *Staphylococcus*, and an evident lack of strictly anaerobic bacteria. In contrast, the advanced clinical stages mainly comprise strictly anaerobic bacteria.

Interestingly, uncommon species detected in other habitats were isolated, such as *Aneurinibacillus aneurinilyticus*, *Pseudomonas oryzihabitants,* or *Trueperella pyogenes* (Table 2).

### 3.4. Microbiota Composition Diversity

Even when the isolate frequency was similar among both groups, there were differences regarding the specific diversity composition. The alpha index profile shows a major diversity at the genera level for the CC group and could be influenced by treatment and CC stages (Figure 3A).

The isolates in the NC group comprised 28 different genera, and 44 genera were identified in the CC microbiota. Notably, the NC group presented more homogeneity or dominance of specific species, probably *Lactobacillus*; in contrast, the CC group presented more genera abundance and richness. Therefore, the evenness value can indicate the constant presence of rare and unique species and considered them as transitory or opportunistic species for both groups (Figure 3(A2)).

According to the rarefaction analysis, the curves overlap (match) in almost 60 isolates, indicating that the remaining isolates from the NC group (*n* = 102) or the CC group (*n* = 116) are distinctively different and specific. In concordance, the Whitaker index (0.41667) indicates the shift among microbiota due to substitution or the presence of other species in the CC group (Figure 3B).

### 3.5. Relevant Genera Describe Specific Microbiota Composition

A multivariate analysis was performed to determine the potential role of each cultured microbiota. Figure 4A shows the most relevant genera for each group; for the NC group, the dominance and homogeneity are for the *Lactobacillus* genus (score_1_ = 35.655), with the lowest score for *Bifidobacterium* and *Gardnerella* (score_1_ = 4.671). For the CC group, it is mainly *Streptococcus* (score_1_ = 20.702), and the lowest score was for *Candida* (score = 6.387) and *Bacteroides* (score_1_ = 4.163). A constant for both groups is the presence of *Staphylococcus* (score_1_ = 35.887), *Corynebacterium* (score_2_ = 11.191), and *Escherichia* (score_1_ = 10.347) named as SEC by its initials and *Enterococcus*, *Peptoniphulus,* and *Prevotella* at a lower frequency. Then, we grouped all these genera with SEC in a cluster called “scavengers”, comprising all the genera preserved despite the changes in microbiota composition or epithelial transformation (Figure 4A).

The hierarchical cluster analysis shows 7 women microbiota clusters (Figure 4B, A–G). Generally, many transitory and opportunistic genera were observed for both groups (Figure 4B). Cluster A is characterized mainly by *Lactobacillus* dominance in the NC women, while Cluster B comprises *Lactobacillus* dominance along with *Staphylococcus* presence. It is vital to observe the presence of *Candida* in the CC women, which indicates that the yeast could play a role in the cancer progression or transition. Cluster C shows that another event for CC patients could be related to the presence of anaerobes genera.

Finally, SEC (Figure 4B) dominates D–G clusters. The isolation of four bacteria strains was remarkable, whose identification was non-conclusive, so they could be novel species.

## 4. Discussion

The traditional bacteria culture strategy for cervical microbiota allowed for the identification and isolation of 43 bacteria and 1 yeast genera previously reported in the repertoire of the vaginal microbiome [18,19]. However, the isolations obtained in this pilot work represent an opportunity area for the analysis of in vitro associations and interactions using active and pure microorganisms obtained from the microenvironment [20]. The above may be one of the advantages of culturomics above metagenomics studies, despite all the advantages these new strategies represent.

Non-autochthonous bacteria were detected in the CC patients at advanced disease stages (IIB-IVA); these species could represent opportunistic pathogens in immunocompromised patients, which are more susceptible to acquiring new infections during cancer treatment, as it has been described in patients admitted to the hospital during their therapy [21]. These changes are also associated with intimate homemade unorthodox habits derived from miraculous remedies or desperate strategies for better prognosis prospects [22,23], a fact that needs special attention.

In the present study, the distribution of genera isolated from NC patients was grouped into four dominant phyla in order of abundance, corresponding to Firmicutes, and followed by Actinobacteria, Proteobacteria, and Bacteroidetes. This distribution has been reported previously in taxonogenomics descriptions [19]. As mentioned, most isolates detected in the NC group are from the Firmicutes phylum; this is consistent with several reports about the higher relative abundance of bacterial species from the Firmicutes phylum belonging to the genus *Lactobacillus* in the healthy female vaginal tract [24].

Moreover, our findings could be supported by the already published papers focused on massive sequencing. Hence, the enrichment of Proteobacteria and Firmicutes phyla observed for the CC group has been reported in other types of cancer, including head and neck [25], pancreas [25], and in CC [25]; also, DNA from bacteria of these groups has been detected in several tumor types, with different mechanisms of oncogenesis participation previously described [26]. Thus, in neoplastic processes, bacterial genera belonging to Firmicutes and Proteobacteria phyla could be expected.

One of the most critical factors associated with cervical microbiota challenges was dysbiosis, a condition characterized by a depletion of *Lactobacillus* and the dominance of a different genus, which could be the result of the enhancement of different variables affecting the course of carcinogenesis; the diversity index, the abundance, and enrichment of anaerobes genera in the CC group evidenced this alteration in this work. The vaginal tract dysbiosis mechanism is widely supported by previous cervical and vaginal syndromes and infection data analyses already published [27,28]. The culturomics approach contributes to the knowledge of the living microbiota in some stages of cervical carcinogenesis and discovers valuable contributions scarcely studied and published.

It is essential to mention that, even when some NC women are clustered with the CC group, the data support a constant presence of anaerobes and opportunistic genera, and these could be the prelude to other infections, as has been described for HIV or HPV infections [29,30,31]. Further, this could support an opportunity area for a therapeutic approach by probiotics leading to a healthy vaginal environment [32,33].

In agreement with previous metagenomics reports that concluded that the Latina vaginal and cervical microbiome is more diverse than ones in other female populations [1], the results support a higher diversity among patients without visible symptoms of infection or cancer; moreover, the increased diversity in the CC microbiota compared to the NC microbiota is notable, the anaerobic bacteria being the enriched genera, and *Bacteroides* the most common isolated genus, an appealing future study target. Previous works reported the risen-on anaerobes genera under stress conditions, so any factor that triggers any adverse condition can lead to dysbiosis if the surrounding microenvironment is favorable [27,31]. Furthermore, each patient’s immune response could promote self-maintenance, adaptation to local hypoxia, and prompt a bad prognosis [34,35]. The anaerobes could be potential CC hallmarks and an opportunity area for research; the scientific community must consider both external treatment (radiotherapy) or internal (brachytherapy), as well as the cancer stage because it could result in different stress level conditions for each cell type studied.

The role of *Lactobacillus* in the CC group may require special attention because this genus is absent in most CC women.

The yeast is fully recognized in different human fungal opportunistic infections because of its adaptability through genome plasticity, which leads to a more significant response capacity to stress situations and possible pathogenicity mechanisms [36]. Its role in premalignant and malignant oral and cervical mucosa lesions has been documented [37,38]. This finding suggests that the immunocompromised patient could develop a favorable microenvironment for the occurrence of other opportunistic or pathogenic organisms as it has been described in diverse intestine conditions [39]; also, the stress allows mucosal vulvovaginal tissue inflammation and opportunistic genera surfacing [40].

Interestingly, recent studies in the carcinogenesis process remark the widely known importance of normal microbiota; for example, *Staphylococcus* has a dual role in different types of cancer to favor patients’ bad prognosis [41]. In addition, *Enterococcus* and *Streptococcus* have been considered as biomarkers in different types of cancer such as gastric or colorectal cancer [42,43]. Meanwhile, *Peptoniphulus* was reported in a medical sepsis case, where it was isolated from the blood of a CC patient during chemotherapy treatment and before invasive procedures; this indicates that anaerobes are important in this type of patient due to their potential as opportunistic genera or by playing a role as protagonist microorganisms [44]. Our findings show that scavengers were present on the cervical epithelium, with or without cancer. All these bacterial genera can adapt to stress conditions, avoiding the immune response [45]. We hypothesize that these native endogenous microorganisms, such as anaerobes, scavengers, and opportunistic and lactic acid bacilli, are essential for maintaining cervical homeostasis. The probable selfish cooperation between these microorganisms with others, collaborating unconsciously, unspecifically, and bi-directionally in a *Ying–Yang* equilibrium process, could permit them to act as scavengers or protagonists/antagonists under specific circumstances: they will be collaborating or in an inactive state (surviving) under lactic acid bacilli presence, so the same bacteria group has an antagonist role under stress conditions as an opportunist; they favor the entrance of others participants that have a greater influence on changes at the epithelium level, such as HPV [4,31,46,47], and also act as a protagonist after these changes as a promoter of the ideal carcinogenesis microenvironment occur, changing the chemical communication among bacteria and the cervical epithelium, and favoring the ecological microprogression from *Lactobacillus* to anaerobe predominance. The model highlights the importance of implementing mechanistic studies to elucidate the role of the microorganisms’ microprogression during cancer transition or progression pathways and the possible association among all microorganisms in the microbiota as microenvironmental mediators for a successful HPV infection. Meanwhile, the data trend shows the importance of considering the external and internal therapy influences as stress mediators and the increased diversity consequences, showcasing probiotics as a complementary therapy to restore the “normal” balance. Our proposed model is shown in Figure 5.

## 5. Conclusions

The living and cultured cervical microbiota, with or without cancerous cells, is attributable to Firmicutes and Proteobacteria. A vast group of bacteria could act as scavengers by maintaining the cervical conditions in a *Ying–Yang* process, supporting a slow microprogression or shift phenomena from a microenvironment enriched with lactic bacteria to anaerobes and opportunistic genera. In this phase, the role of *Candida* and anaerobes could favor opportunistic bacteria–bacteria and bacteria–cells intercommunication. Here, we propose a transitional state in which microbial composition and dynamics may be essential for maintaining the balance of the cervical microenvironment or causing its disruption.

## Figures and Tables

**Figure 1 cancers-16-00314-f001:**
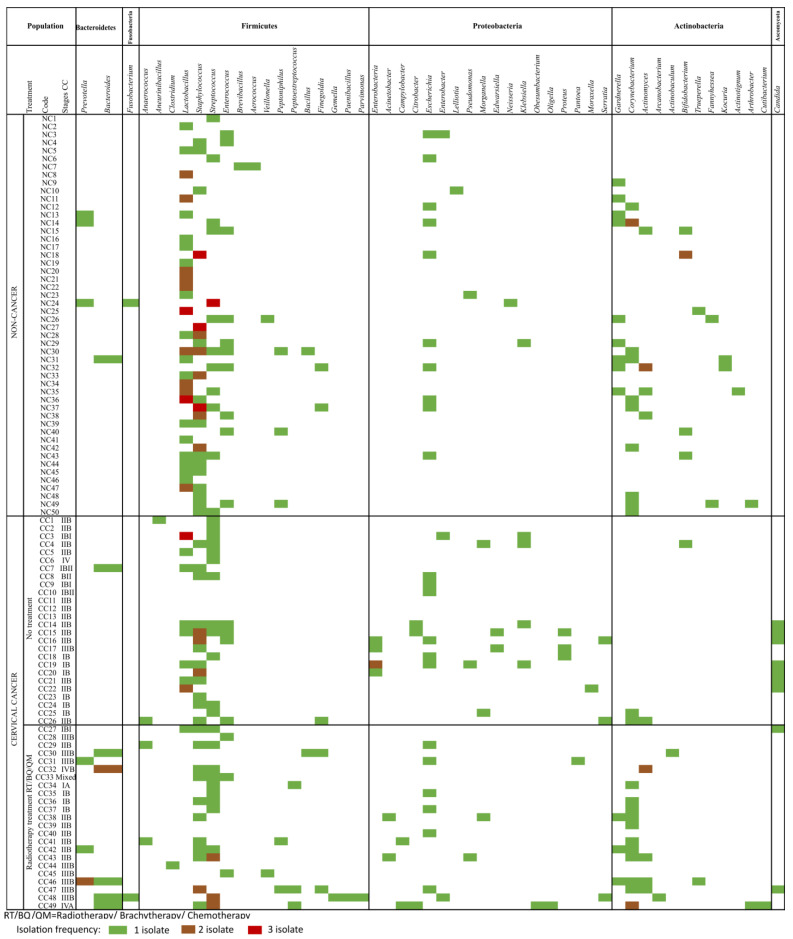
Cervicovaginal microbiota detected in cytology samples taken from non-cancer and cervical cancer patients.

**Figure 2 cancers-16-00314-f002:**
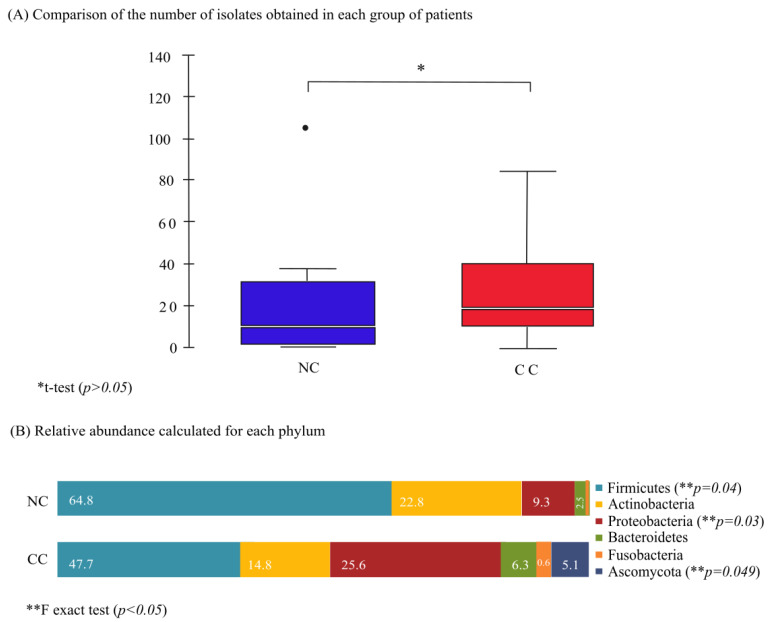
Differences in relative phyla abundance between non-cancer and cervical cancer groups. (**A**) The box plots compare the mean number of isolations obtained for NC and CC populations; despite the difference in the identity of isolates for each group, the * *p*-value higher than 0.05 was considered insignificant. (**B**) Cervical microbiota was sorted in phyla taxa; the average relative abundances of each phylum were represented on the bar chart as an accumulative percentage, and to determine the predilection of each phylum for a group of patients, non-cancer or with cervical cancer, Fisher’s exact test was used (** *p* < 0.05). Firmicutes and Fusobacteria phyla were the most abundant in both groups; meanwhile, Proteobacteria and the other phyla were more significant in the CC group.

**Figure 3 cancers-16-00314-f003:**
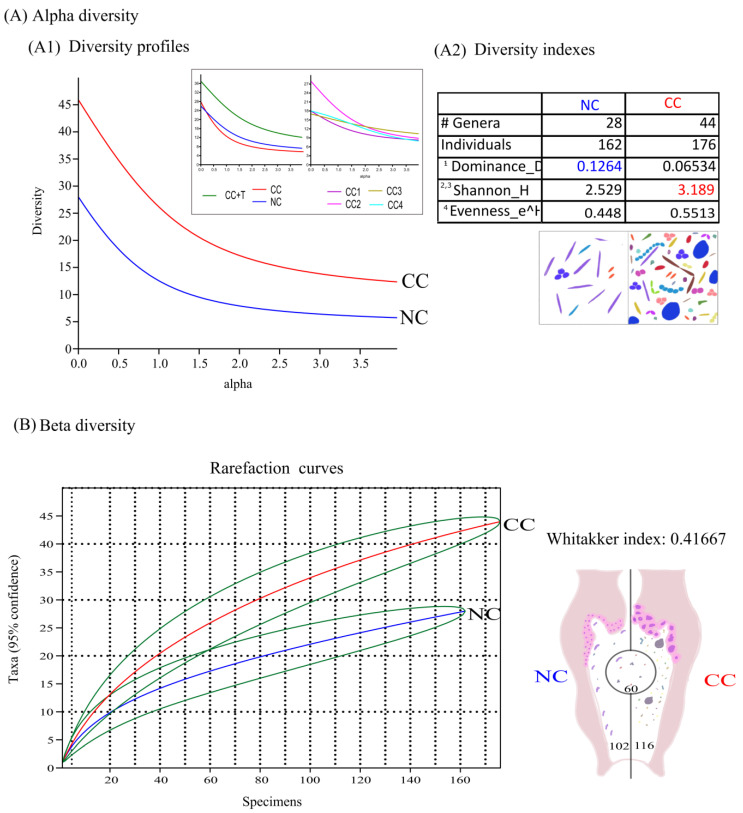
Microbiota diversity comparison between non-cancer and cervical cancer groups of women. The graphs show the results for alpha and beta diversities. (**A**) Alpha diversity profiles. (**A1**) show that non-cancer microbiota (NC blue line) have more specific genus 1 dominance, whereas microbiota from the CC group (Red line) are more diverse. The upper square shows the diversity increase attributable to treatment (CC + T green line) and cancer Bethesda stages (CC1 purple line, CC2 pink line, CC3 yellow line and CC4 aqua line). (**A2**) Diversity indexes, the table show there is more 2 richness and 3 abundance in the number of genera on CC (red value), meanwhile NC has more 1 dominance (blue value), however, both groups have similar 4 evenness or transitory microbiota presence, the graph under the table exemplify pictorially the diversity indexes meaning for NC (left) (dominance index) and CC (right) (richness and abundance indexes), using different microbial morphologies examples (different color and shapes). (**B**) Beta diversity indicates the changes in the genera composition in proportions of 0.41 according to the Whittaker index and rarefaction curves, NC (blue line) and CC (red line) with 95% confidence intervals (green lines); the graph at right is useful to illustrate that NC (blue) and CC (red) microecosystems are distinct in microbiota populations, despite that initially they shared some core genera.

**Figure 4 cancers-16-00314-f004:**
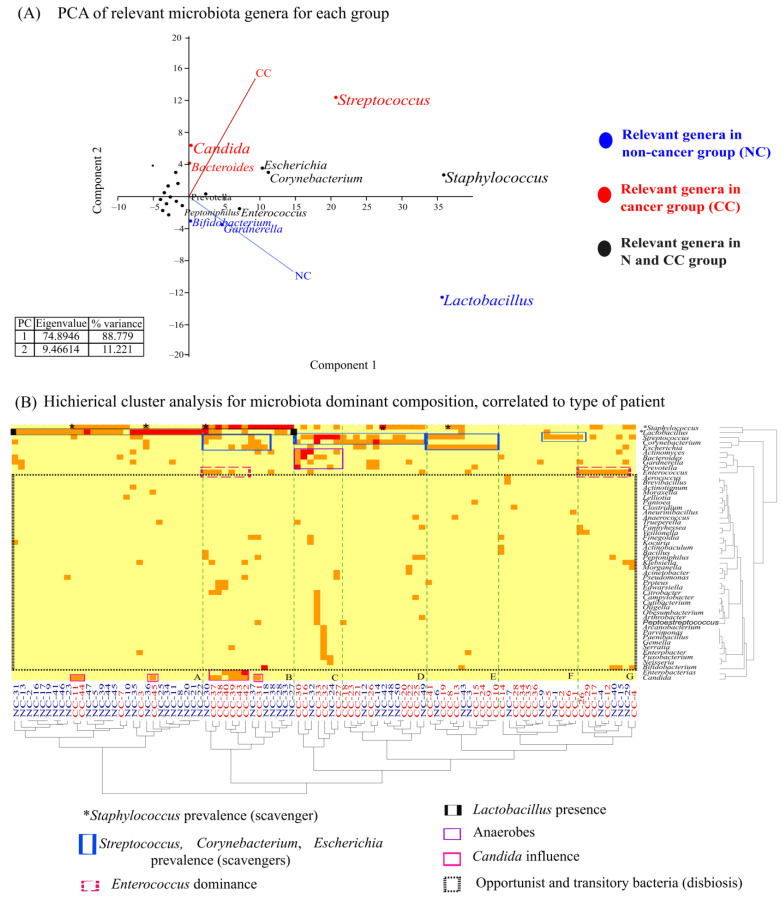
Relevant genera of the microbiota identified in non-cancer and cervical cancer groups. (**A**) PCA plot shows that the relevant genera in microbiota from the non-cancer group are *Lactobacillus*, *Gardnerella,* and *Bifidobacterium*; meanwhile, *Streptococcus*, *Candida,* and *Bacteroides* are more relevant for the CC group, although, *Staphylococcus*, *Corynebacterium*, *Escherichia, Peptoniphulus, Prevotella,* and *Enterococcus* are relevant to both groups. (**B**) Hierarchical clustering analysis for both populations generate seven clusters (A–G) corresponding to isolation frequency correlations among microbiota genera. Clusters A–B had *Lactobacillus* dominance, mainly from non-cancer women (NC); meanwhile, clusters C–G dominated CC women samples. The more relevant bacteria for each cluster were included in blocks of color; we observed four principal characteristics: the dominance of *Lactobacillus* genera in the NC group, the constant presence of opportunistic and transitory genera, the possible influence of some specific genus on CC microbiota, and the permanence or prevalence of other specific genera in both groups, named here for illustrative purposes as “scavengers”.

**Figure 5 cancers-16-00314-f005:**
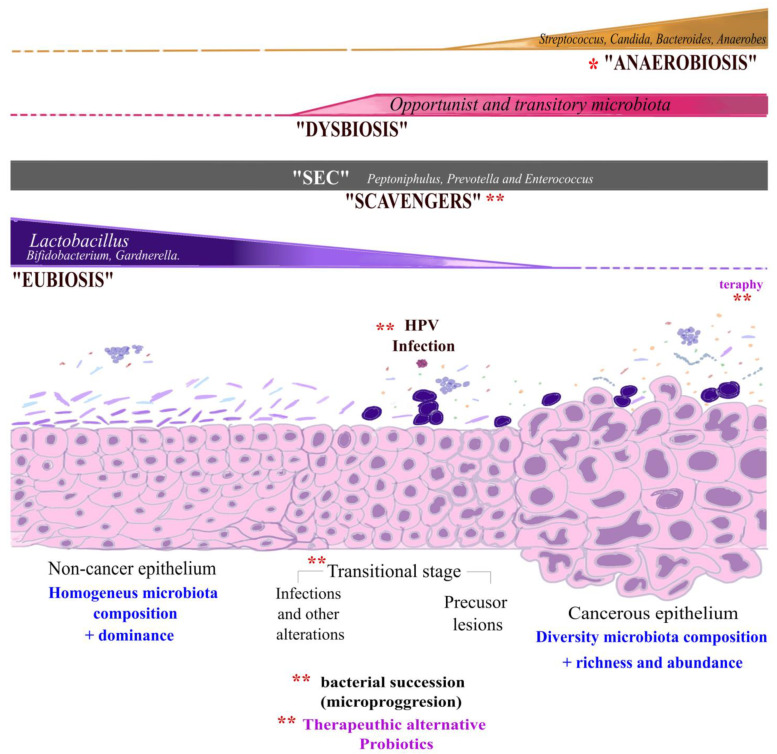
Distribution of cervical microbiota in study populations. The image shows the transformation process from normal cervical epithelium to invasive carcinoma and the microbiota composition associated with each stage. Initially, the “Eubiosis” condition on healthy epithelium was characterized by the dominance of *Lactobacillus* spp. and other lactic acid bacteria. However, the depletion of these bacteria caused a dysbiosis condition characterized by increased diversity of opportunistic and transitory microbiota established during cervical disease progression. We proposed the * “Anaerobiosis” status in cancerous cells because of a noticeable increase in anaerobic bacteria. Here, it is proposed that the role of *Staphylococcus*, *Escherichia*, and *Corynebacterium* (SEC) added to *Peptoniphulus*, *Prevotella,* and *Enterococcus* genera, together denominated as “Scavengers”, have constant participation in non-cancer and cancer populations. The transitional stage could be an opportunity for a therapeutic approach with probiotics. ** Shows the possible areas for implementing mechanistic studies to understand and improve the model.

**Table 1 cancers-16-00314-t001:** Demographic and clinical features of non-cancer and cervical cancer patients.

Characteristics		Non-Cancer (*n* = 50)	Cervical Cancer (*n* = 49)	*p*-Value
Age (years)	Mean	40.62	51.2	*p =* 0.2489
	<40	19	9	*p =* 0.0588
	>40	31	40	*p =* 0.2855
Body Mass Index	<25	20	36	***p =* 0.0325**
	>25	30	13	***p =* 0.00095**
First Sexual Intercourse (years)	<18	0	29	***p* < 0.001**
	>18	50	20	***p =* 0.0003**
Number of sexual partners	<2	26	30	*p =* 0.5930
	>2	24	19	*p =* 0.4458
Smoking	Yes	13	15	*p =* 0.7055
	No	37	34	*p =* 0.7218
* Nugent score	<3	48	Nd	Nd
	>4	2	Nd	Nd
** Amsel criteria	Positive	24	8	***p =* 0.0047**
	Negative	26	41	***p =* 0.0669**
HPV test	Positive	0	39	***p* < 0.0001**
	Negative	50	10	***p* < 0.0001**
Diagnosis	I-II	0	38	***p* < 0.001**
	III-IV	0	11	***p =* 0.0009**

* Presence of *Lactobacillus*; ** Clue cells on wet mount microscopy; FIGO Clinical stage CC.

**Table 2 cancers-16-00314-t002:** Uncommon bacterial isolates detected in cervical cancer (CC) and non-cancer (NC) groups.

Phylum	Number of Isolates		Detection in Other Habitats
	NC	CC	
Firmicutes			
*Aneurinibacillus aneurinilyticus*		1	Liquor [15], marine sediment [16]
*Bacillus* spp.	1		Moldy corn [17], soil Himalayan [18]
*B. amylolicuefasciens*		1	
*Paenibacillus* spp.		1	Cosmopolitan [19]
Proteobacteria			
*Obesumbacterium proteus*		1	Breweries [20]
*Pantoea agglomerans*		1	Soil, insect, phytopathogenic [21]
*Pseudomonas* spp.	1		Plant [22], bath Sponge [23]
*P. oryzihabitans*		1	
*Edwarsiella tarda*			Marine animals [24]
Actinobacteria			
*Truperella pyogenes*		2	Domestic and wild animals [25]

## Data Availability

The data are not publicly available due to restrictions of privacy and ethics.

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
