# Peer review of "Cultivable Microbiome Approach Applied to Cervical Cancer Exploration"

_cancers, 2024, doi:10.3390/cancers16020314_

Round 1
Reviewer 1 Report
Comments and Suggestions for Authors
In this study, Mulato-Briones et al. studied the microbiome composition of women with cervical cancer and healthy individuals using the culturomic approach. The results showed a difference in the composition of the microbiota between cervical cancer patients and non-cancer people. The topic of this study is meaningful, but I have a few concerns. Here are some comments on this paper:
1. There is no line number in the paper, which makes it hard to point out the issues.
2. Names of bacterial phylum should not be italicized, such as in the abstract “Firmicutes and Proteobacteria”. Please revise them throughout the paper.
3. The introduction section is short, and it is recommended to add descriptions of culturomic approach.
4. Section 2.6 the information about “16SF/16SR” was lacking and title should be “Bacteria 16S rDNA identification”.
5. Section 2.8 should be “A p value of less than 0.05 is considered significant.”
6. Section 2.9 “alfa” should be “alpha”.
7. Section 3.1, it is proposed to add content about Amsel criteria.
8. Figure 2 A, “*p=0.913” p value higher than 0.05 was insignificant.
9. Table 2 did not show the results clearly and was difficult to read. It was suggested to revise it.
10. Through Figure 1 and as mentioned by the authors in section 3.4 “Even when the isolate frequency was similar among both groups, there were differences regarding the specific diversity composition.” Did the NC1 sample contain only one Streptococcs and the NC2 sample contain only Lactobacillus? If that's true, it means that the results of the table for diversity calculations are very sparse, and I consider that it affects the results a lot and could the authors provide any explanation for this?
11. In conclusion, what is ““Ying-Yang” process”?
Reviewer 2 Report
Comments and Suggestions for Authors
The authors present a study that confirms the differences in cervico-vaginal microbiome between cervical cancer and non cancer women using culturomics as opposed to high throughput genomic approaches. There has been significant research conducted in this arena over the past decade and this study adds to the body of evidence for a role of the microbiome in cervical cancer. Below are my specific comments;
1. Can you provide some statistics on the power of your study given the limited sample size for the two main categories of analysis - CC v/s NC? While statistically significant differences are relevant for the two broad categories that you are comparing, there is significant stratification within the samples, and any differences seen for sub categories may not be strong enough.
2. If culturomics is the key differentiator for your study compared to the large set of papers published in this area, then this aspect needs to be called out better in the discussion. Otherwise it is yet another study that repeats the conclusions from all other studies.
3. The model presented in the discussion section is more of an "association" description rather than a hypothesis of how HPV infection might impact a change in the microbiota - that is, is the change in microbiome characteristics a cause or result of HPV infection and subsequent cervical cancer progression? Perhaps recommendations on what points in this model could one design mechanistic studies to elucidate progression pathways would be a good use of this model.
4. For this study to add value to the scientific knowledge about the cervico-vaginal microbiome and its role in cervical cancer, can you call out the unique contributions you have made using the culturomics approach?
Comments on the Quality of English LanguageSome sentences in the abstract and manuscript are awkward. A thorough review would be beneficial.
Reviewer 3 Report
Comments and Suggestions for Authors
Works about microbiota differences in cervical cancer/non-cancer women are available for some time now. Here the authors tried a modern approach with Vitek mass spectrometry and rDNA sequencing. This is a good idea which may prove useful results for the future study of cervical cancer. However I find one big weakness in the study protocol, the homogeneity of the cervical cancer population. In fact the 49 patients batch referred from Brachytherapy and Radiology units is too heterogeneous in terms of cervical cancer treatment and stage. It is known that radiotherapy and chemotherapy alter the microbiota profile in different ways. This may in part explain the alpha index profile and Amsel criteria for cervical cancer group. Brachytherapy, as an invasive treatment method can favor infections with Streptococcus, Candida as pointed out by the authors themselves in the Discussion paragraph. The cervical cancer group must be divided in pts undergoing chemo/rt at the time of cervical sampling, pts at a specific time interval after the completion of chemo/rt and cervical cancer pts that had yet not received cancer treatment at the time of cervical sampling.
Also, some minor issues:
- abstract: last line needs to be rephrased for better clarity
- results: "Interestingly, the CC population in FSI...". I suggest replacing "interestingly" with " as expected"
Comments on the Quality of English LanguageMinor English corrections are needed such as "a piece" or "duplicate tested" in paragraph 2.5.
Round 2
Reviewer 1 Report
Comments and Suggestions for Authors
I appreciate the authors’ responses! After carefully reviewing the authors’ response and the revised manuscript. I have no further concerns.
Reviewer 2 Report
Comments and Suggestions for Authors
I would recommend including the information provided in the author response to my question about the statistical analysis, in the "statistical analysis" section of the manuscript.
Comments on the Quality of English LanguageMinor editing on some of the new verbiage in the manuscript needed.
Reviewer 3 Report
Comments and Suggestions for Authors
Your work now looks considerably better. I still suggest that some stratification should exist among CC patients, at least in the form of irradiated vs non-irradiated patients, chemo vs no-chemo which should not be that hard, especially considering the batch is just 49 strong. However, I did understand your point and I won't hold it against publishing this paper.
Good work !
